# Pilot Study by Liquid Biopsy in Gastrointestinal Stromal Tumors: Analysis of *PDGFRA* D842V Mutation and Hypermethylation of *SEPT9* Presence by Digital Droplet PCR

**DOI:** 10.3390/ijms25126783

**Published:** 2024-06-20

**Authors:** Rocío Olivera-Salazar, Gabriel Salcedo Cabañas, Luz Vega-Clemente, David Alonso-Martín, Víctor Manuel Castellano Megías, Peter Volward, Damián García-Olmo, Mariano García-Arranz

**Affiliations:** 1New Therapies Laboratory, Health Research Institute-Fundación Jiménez Díaz University Hospital (IIS-FJD), Avda. Reyes Católicos, 2, 28040 Madrid, Spain; luz.vega@quironsalud.es (L.V.-C.); david.alonsom@quironsalud.es (D.A.-M.); damian.garcia@quironsalud.es (D.G.-O.); mariano.garcia@quironsalud.es (M.G.-A.); 2Surgeon Esophagogastric Unit, Hospital Fundación Jiménez Díaz, Avda. Reyes Católicos, 2, 28040 Madrid, Spain; gabriel.salcedo@quironsalud.es (G.S.C.); pwvorwald@quironsalud.es (P.V.); 3Anatomical Pathology Service, Hospital Fundación Jiménez Díaz, Avda. Reyes Católicos, 2, 28040 Madrid, Spain; victor.castellano@quironsalud.es; 4Department of Surgery, Fundación Jiménez Díaz University Hospital (FJD), 28040 Madrid, Spain; 5Department of Surgery, Universidad Autónoma de Madrid, 28034 Madrid, Spain

**Keywords:** rare disease, gastrointestinal stromal tumor (GIST), biomarkers, tissue biopsy, liquid biopsy, early diagnosis, cell-free DNA, droplet digital PCR (ddPCR), *PDGFRA*, hypermethylated *SEPT9* gene

## Abstract

Tissue biopsy remains the standard for diagnosing gastrointestinal stromal tumors (GISTs), although liquid biopsy is emerging as a promising alternative in oncology. In this pilot study, we advocate for droplet digital PCR (ddPCR) to diagnose GIST in tissue samples and explore its potential for early diagnosis via liquid biopsy, focusing on the *PDGFRA* D842V mutation and *SEPT9* hypermethylated gene. We utilized ddPCR to analyze the predominant *PDGFRA* mutation (D842V) in surgical tissue samples from 15 GIST patients, correlating with pathologists’ diagnoses. We expanded our analysis to plasma samples to compare DNA alterations between tumor tissue and plasma, also investigating *SEPT9* gene hypermethylation. We successfully detected the *PDGFRA* D842V mutation in GIST tissues by ddPCR. Despite various protocols to enhance mutation detection in early-stage disease, it remained challenging, likely due to the low concentration of DNA in plasma samples. Additionally, the results of Area Under the Curve (AUC) for the hypermethylated *SEPT9* gene, analyzing concentration, ratio, and abundance were 0.74 (95% Confidence Interval (CI): 0.52 to 0.97), 0.77 (95% CI: 0.56 to 0.98), and 0.79 (95% CI: 0.59 to 0.99), respectively. As a rare disease, the early detection of GIST through such biomarkers is particularly crucial, offering significant potential to improve patient outcomes.

## 1. Introduction

Gastrointestinal stromal tumors (GISTs) are the most common type of stromal tumor of the digestive tract, with an incidence of 7–15 cases per million. The disease affects adults aged 50–70 years with no gender predilection [1]. It is believed that GISTs came from the interstitial cells of Cajal, which are the pacemaker cells of gastrointestinal movement [2,3]. The most common organs affected by GISTs are those within the gastrointestinal tract, including the stomach (50%), small intestine (25%), rectum (5%), and esophagus (<5%) [4]. The biological behavior of a GIST varies, where the first line of curative treatment is surgical resection [1].

Most GISTs (80–90%) are due to mutations in two genes that code for receptor tyrosine kinases: these kinases are *KIT* and the platelet-derived growth factor receptor alpha gene (*PDGFRA*) [1,5,6,7,8,9]. Around 70% of GISTs have mutations in the *KIT* gene, mostly in exon 11, although they can also be found in exons 9, 13 and 17 [10]. Mutations in the *PDGFRA* gene occur in 5–7% of GISTs and mainly affect exon 18 (~5%), or more rarely exon 12 (~1%) and exon 14 (<1%) [11]. Oncogenic activation of *KIT* or *PDGFRA* serves to track GIST progression at all stages of the disease. The *KIT*/*PDGFRA* genotype predicts response to imatinib as first-line therapy and to other tyrosine kinase inhibitors (TKIs) if imatinib fails [4,5,10,12,13]. Nevertheless, despite a clinical benefit rate of 80%, most GIST patients experience disease progression after 2–3 years of imatinib therapy [14].

Currently, tissue biopsy is the most widely used method for accurate diagnosis of GISTs; however, this approach has several limitations: it is invasive, does not represent the heterogeneity of the tumor, multiple tests over time are not always possible, and it represents the tumor status at a specific point in time. On the other hand, liquid biopsy allows easy repeat testing, real-time monitoring, represents the heterogeneity of the tumor and therefore represents a dynamic disease state, as well as being cheaper and more easily reproducible [15].

Given that TKIs are only effective against a subset of tumors with *KIT* or *PDGFRA* mutations in GISTs, close monitoring of tumor dynamics with non-invasive methods such as liquid biopsy is necessary. Although there are several components in peripheral blood, most recent liquid biopsy studies have focused on circulating tumor DNA given the stability of the DNA itself [12]. The identification of mutations in the *KIT* and *PDGFRA* genes in GISTs allows the selection of patients for targeted therapies. It is important to have a tool for early diagnosis, improved prognosis, real-time monitoring of the disease, and ultimately to improve the survival rate [15].

For the cancer care, the molecular testing is crucial, especially at diagnosis [16]. Molecular diagnostic testing with relevant predictive biomarkers such as *KIT* and *PDGFRA* is becoming a routine clinical decision-making practice [17]. A variety of methods are used for mutation detection, including Sanger sequencing, pyrosequencing, and next-generation sequencing (NGS); however, these methods are expensive, slow, require amounts of DNA larger than 100 ng, and in some cases, there is not enough tumor material available for molecular testing. Alternative methods for mutation detection, ideally non-invasive and allowing analysis with less DNA, are therefore needed. Recent advances in molecular pathology therefore allow the detection of tumor-specific mutations in circulating tumor DNA (ctDNA) extracted from blood plasma (liquid biopsy) [17,18], positioning colorectal cancer as the ideal target for liquid biopsy (46% ctDNA detection in early-stage, 73% of stage II–III cases, and 90% of patients with localized and metastatic CRC) in digestive diseases [19]. Liquid biopsy has entered the era of personalized medicine and with regard to GISTs, a global effort to translate the use of liquid biopsy to the clinic should be considered mandatory [15].

Since the amount of circulating DNA in the plasma of GIST patients is expected to be small [10], in this study, we decided to analyze the most frequent *PDGFRA* mutations in tissue in our patient cohort by liquid biopsy using droplet digital PCR (ddPCR), as it is more sensitive and accurate than other methods and is particularly useful to detect mutations when there is a shortage of sample [20,21,22,23].

Although the use of ddPCR for *PDGFRA* mutation in GIST cell-free DNA (cfDNA) has been described [17,24] and considering that the amount of cfDNA is proportional to disease progression [25], it remains a challenge as an early diagnostic tool, especially considering the wide variety of primary mutations [12]. For this reason, we have considered analyzing the hypermethylation in the *SEPT9* gene to be a common epigenetic alteration in gastrointestinal cancers [26,27].

The aim of this pilot study was to compare the presence of the most frequent *PDGFRA* mutation (D842V) in tumor tissue within our study population at Hospital Jiménez Díaz, utilizing liquid biopsy via ddPCR. This comparison aims to develop a diagnostic tool for early detection and monitoring of GIST disease in patients. Concurrently, plasma samples from patients underwent ddPCR analysis to detect *SEPT9* hypermethylation, serving as an alternative liquid biopsy tool for the early diagnosis of GISTs.

## 2. Results

### 2.1. Patient and Sample Characteristics

Eighteen plasma samples were obtained upon diagnosis from patients presenting with gastric tumors suspected of GIST pathology at the Surgery Esophagogastric Unit, Hospital Jiménez Díaz (Table 1). Among these, three samples exhibited pathologies other than GISTs (G001, G004, and G011) and were excluded from the study. Notably, all GIST plasma samples were derived from patients with localized illness and no metastasis. The average age was 65 years. Of the 15 GIST plasma samples analyzed, *n* = 8 (53.33%) were collected from male patients, while *n* = 7 (46.67%) were obtained from female patients. The most prevalent DNA alteration occurred in the *KIT* gene, specifically in exon 11 with *n* = 8 (53.33%), followed by alterations in *PDGFRA* exon 18 with *n* = 6 (40%). However, the mutation most frequently observed, hence chosen for PCR probes, was in the *PDGFRA* gene, exon 18 (D842V). The most repeated tumor stage was pT2N0, *n* = 8 (53.33%).

### 2.2. Detection of PDGFRA D842V Mutation in GIST Tissue Tumors Using ddPCR

The pathologist at Hospital Jiménez Díaz provided five coded DNAs from GIST tissues (Table 2). The results indicate that four of these DNAs were mutated according to ddPCR analysis, while one was not. Subsequent comparison with pathologist data revealed that ddPCR achieved a 100% detection rate (Figure 1). Furthermore, multiple dilutions of DNA extracted from GIST tissue were performed to establish the minimum dilution (0.1 ng) necessary for mutation detection within the tissue (Figure 2).

### 2.3. Detection of PDGFRA D842V Mutation in Plasma from GIST Patients by ddPCR

Initially, we examined GIST plasma samples obtained from positive tissues using a conventional protocol involving 3.5 mL of plasma. The average DNA yield isolated from the plasma was 1.35 ng/µL ± 0.87. Despite these conditions, we were unable to detect the *PDGFRA* D842V mutation in plasma samples from GIST patients (Figure 3).

We used the number of *PDGFRA* wild-type (WT) copies as a reference to assess the adequacy of the loaded DNA, then normalized the number of mutated copies accordingly. We observed that the WT sequence in plasma had a lower concentration compared to tissue (108.66 ng/µL vs. 510 ng/µL). This discrepancy may be due to the lower amount of DNA loaded in plasma (8.8 µL × 1.35 = 12 ng) compared to the 100 ng loaded in tissue. Based on this observation, we investigated whether the absence of the *PDGFRA* D842V mutation in plasma was due to the sample itself or to the low DNA concentration loaded. To elucidate this, we proposed three modifications: using fresh plasma, concentrating DNA from plasma through centrifugation or speed vacuum, and increasing the volume of plasma to 12 mL from the original 3.5 mL. The results indicated that using fresh plasma and increasing the volume were the most effective options for obtaining more events in ddPCR aiming to detect minimal residual mutations. However, despite increasing DNA concentration through plasma concentration, we were still unable to detect the *PDGFRA* D842V mutation in patients with confirmed tissue mutations (Table 3).

### 2.4. Detection of SEPT9 Hypermethylation in GIST Tissue by ddPCR

Based on the above results, we decided to test the same DNA GIST tissues using *SEPT9* hypermethylation probes to find out if these tumors were positive for this biomarker. We found that all tissues were positive for *SEPT9* hypermethylation (Figure 4).

### 2.5. Detection of SEPT9 Hypermethylation in Plasma from GIST Patients by ddPCR

Once the presence of hypermethylation of the *SEPT9* gene in GIST tissue was confirmed, we proceeded to analyze it in plasma from GIST patients. Since hypermethylation is an epigenetic process, we analyzed all the plasmas of the 18 patients included in this study, as it is independent of specific mutations. It was found that it is possible to detect *SEPT9* gene methylation in plasmas of GIST patients (Figure 5). No significant differences in *SEPT9* gene hypermethylation were found comparing males and females *p*-value, *t*-test, α = 0.05: 0.2034.

An analysis of the results was performed using the Area Under the Curve (AUC) of the Receiver Operating Characteristic (ROC) [28] for the concentration, ratio, and abundance of hypermethylation of the *SEPT9* gene in healthy subjects and those with GIST (87.5% sensibility of the test). The AUC results were 0.74 (95% CI 0.52 to 0.97) for concentration, 0.77 (95% CI 0.56 to 0.98) for ratio, and 0.79 (95% CI 0.59 to 0.99) for abundance, with the latter being the best at distinguishing between healthy and diseased individuals. To find the threshold for hypermethylation of the *SEPT9* gene in plasma, we compared the methylation levels in healthy subjects with the methylation levels in GIST patients and found that the methylation detected in plasma from GIST patients is significantly higher than that detected in healthy subjects, with the ratio and abundance threshold for pathological plasma being 0.015 and 1.5%, respectively (Figure 5).

## 3. Discussion

The incidence of GISTs is approximately 1 per 100,000 individuals per year in most countries. The molecular profile exhibits variability, ranging from *KIT* or *PDGFRA* mutations to rare subtypes. Patients harboring *KIT* or *PDGFRA* mutations demonstrate improved survival rates with tyrosine kinase inhibitors (TKIs) like imatinib. However, resistance to TKIs arises primarily from subclonal cells carrying resistance mutations in *KIT* or *PDGFRA*. While second and third-line treatments exist, uncertainties persist despite these advancements, particularly regarding the role of surgery in advanced disease [29]. In our hospital, the diagnostic techniques for GISTs were endoscopic ultrasound for tumor localization and staging, along with core needle biopsy to confirm the diagnosis of a gastrointestinal stromal tumor. An abdominopelvic computed tomography scan was performed to assess the proximity of the tumor to other adjacent abdominal organs and to rule out metastatic disease. Given that these techniques are more expensive and bothersome for the patient, we considered new diagnostic tools necessary for this type of tumor with the aim of improving the quality of health care. In this regard, we focused on searching for relevant genetic alterations as biomarkers through liquid biopsy using digital droplet PCR (ddPCR).

As indicated by the literature [10], the majority of mutations in GIST pathology occur in the *KIT* gene (70%). In our cohort of GIST patients, we have found that it is also the most altered gene (8/15 patients, 53%). Despite this, due to the lack of homogeneity in *KIT* gene mutations in our samples and considering that specific probes for each mutation must be used through ddPCR, we have chosen to analyze the most recurrent mutation, which is in the *PDGFRA* D842V gene (5/15, 33%). Before analyzing this mutation in the plasma of GIST patients, we validated this study using ddPCR with tissue DNA for diagnosis from our hospital’s blood bank. The results were compared with those of pathological anatomy, and we confirmed that 100% of the cases could be detected through ddPCR. On the other hand, given that ddPCR is much more sensitive and specific than conventional PCR for detecting point mutations [20,21,22,23], we conducted a mutation detection limit assay using tissue DNA, and the results indicate that up to 0.1 ng of tumor DNA can be used using this technique. This is of added value considering that sometimes the tumor specimen is insufficient, or the tumor DNA does not reach the 50–100 ng required by conventional techniques. Therefore, when these conditions are met, ddPCR for the detection of point mutations in GIST can be an alternative technique to aid in the diagnosis of insufficient samples.

Before validating the study with liquid biopsy, the main challenge was the low availability of GIST samples due to its low annual incidence [29]. Another limitation was to include the genetic heterogeneity of GIST tumors, where we had to select the most common mutation in our patient cohort rather than the most altered *KIT* gene. Another challenge was the low concentration of DNA in early-stage GIST plasma samples and the variability in the concentration and quality of cfDNA samples. We had to elucidate why we had not been able to detect the mutation in plasma. Serrano and colleagues indicated that this mutation is not detected in early-stage disease through liquid biopsy and only appears in a few cases of advanced disease. In this study with 18 patients, Serrano and colleagues showed that GIST has relatively low ctDNA shedding and ctDNA was detected only in GIST patients with advanced disease, predicting tumor dynamics in serial monitoring [10]. Here, we set out to determine if the lack of detection was due to factors in the DNA extraction protocol of the sample. Therefore, in our study, we made variations in the volume of mL of plasma used, using fresh plasma instead of cryopreserved, and increasing the concentration of DNA to be used in ddPCR through centrifugation and vacuum. Despite this, we were unable to detect the mutation in plasma, leading us to consider that it might have been due to the low release of tumor DNA into the bloodstream by this type of tumor. Given that GIST is a stromal tumor with a low metastatic incidence (<20%) [10], even lower in our series of patients due to its anatomopathological stage, the concentration of DNA released by tumor cells into the bloodstream was likely very low.

Identifying epigenetic events is emerging as a promising approach for early-stage disease detection, given that the dysregulation of gene expression through abnormal DNA methylation is a well-established phenomenon in tumor biology [30]. For all these reasons and considering the heterogeneity of the molecular profile of GIST, we consider using the analysis of *SEPT9* gene hypermethylation, as it is a very frequent epigenetic alteration in digestive tumors. The advantage of using hypermethylation analysis of the *SEPT9* gene is that it occurs at all stages of carcinogenesis [31], making it easier to detect in early-stage disease. By using this analysis, we overcame the lack of homogeneity of the mutations in the samples and found that, once this alteration was validated in the tissue, we were able to detect it in the plasma of GIST patients. The results of our study, with AUC values of 0.74, 0.77, and 0.79, indicate that measures of *SEPT9* gene hypermethylation have a decent ability to distinguish between healthy subjects and GIST patients. An AUC of 0.79 suggests that the frequency of the hypermethylated *SEPT9* gene is the most effective measure among the three for this distinction. However, due to the wide confidence intervals, increasing the sample size in future studies would be beneficial to obtain more precise and robust estimates. This would also enhance confidence in the statistical significance of the observed results. Although statistical analysis showed significant results when comparing *SEPT9* hypermethylation in healthy subjects with those with a GIST, and the AUC also indicated significance, it is important to note that the limited number of samples, due to the rarity of the disease, is a limitation. To ensure a powerful analysis with statistically significant results, a larger sample size would be needed. Therefore, despite the multiple difficulties presented by this study, especially regarding the number of samples that can be obtained since we are dealing with a rare disease [32,33,34] we offer a new analysis to help diagnose localized GIST disease through liquid biopsy using *SEPT9* gene hypermethylation analysis. Unfortunately, the detection of hypermethylation in the *SEPT9* gene is not specific for GISTs, as different gastrointestinal tumors show elevation. Thus, chronic inflammation is believed to promote carcinogenesis through genetic alterations such as chromosome instability and microsatellite instability, including DNA hypermethylation [35]. However, the relationship between hypermethylation of the *SEPT9* gene and inflammation is still not well studied and it would be interesting to focus future research on this field. Nevertheless, we believe that it is an unaggressive test that may be useful in diagnosis.

Here we have shown that the analysis of *SEPT9* gene hypermethylation by liquid biopsy using ddPCR is a powerful tool to aid in the diagnosis of GISTs. Not only can it be applied to diagnosis, but the variation of hypermethylation ratios after surgery can aid in the assessment of disease prognosis after treatment as well as the assessment of minimal residual disease after treatment as has been studied in other types of digestive tumors such as colorectal cancer. The use of circulating markers currently represents a promising tool in oncology to improve the quality of health care, allowing earlier diagnoses in a non-invasive way.

## 4. Materials and Methods

### 4.1. Sample Collection

This was a prospective, national, single-center, observational cohort study approved by Hospital Fundación Jiménez Díaz (FJD), Madrid, Spain, with registration code: PIC104-22_FJD. We recruited 18 patients diagnosed with localized disease, three of whom were later determined not to have GIST, from the Department of General and Digestive Surgery at Hospital FJD. Patients of both sexes, between 18 and 90 years old, with a suspected GIST were included in this study for the use of plasma samples. The plasma samples from patients suspected of having a GIST were collected at the time of diagnosis and prior to surgery. Patient selection was carried out after diagnosis provided by anatomical pathology at our hospital. Additionally, plasma samples were collected from healthy donors at the hospital’s blood bank, with approval code: PIC137_2017-FJD. The healthy donor selection was randomized, resulting in a participant distribution of 66.7% male and 33.33% female, all aged between 18 and 70 years. Prior to sample collection, signed informed consent was obtained from all participants. Peripheral blood samples were collected from each participant using 2–3 EDTA tubes (BD Vacutainer, Plymouth, UK) to obtain 20–30 mL of whole blood. Plasma was isolated within 2 h of blood collection through two centrifugation steps: first at 1800 g, 4 °C, for 10 min to eliminate cell debris, followed by centrifugation at 3000× *g*, 4 °C, for 10 min. Isolated plasma was then stored frozen at −80 °C until further analysis.

### 4.2. DNA Isolation

#### 4.2.1. GIST Tissue

DNA from GIST tissue was provided by the anatomical pathology department of the hospital using Cobas^®^ DNA Sample Preparation Kit (Roche Diagnostics, Vienna, Austria) for DNA extraction from formalin-fixed, paraffin-embedded (FFPE) tumor tissue sections. DNA quality was measured with a NanoDrop 1000 spectrophotometer (Thermo Fisher Scientific, Waltham, MA, USA) and the DNA concentration was quantified by fluorimetry in a Qubit^®^ 4.0 Fluorometer using the Qubit™ 1X hsDNA Assay Kit (Thermo Fisher Scientific, Eugene, OR, USA).

#### 4.2.2. Plasma Samples

CfDNA extraction from plasma samples was performed using QIAamp Circulating Nucleic Acid Kit (Qiagen, Hilden, Germany) following the manufacturer’s protocol. An amount of 4 mL of plasma was used for cfDNA extraction. Afterward, to increase the DNA concentration, 12 mL of plasma was used. CfDNA isolated was quantified by nanodrop to measure the DNA quality and then the concentration of DNA was measured by Qubic 4.0 with Qubit™ 1X hsDNA Assay Kit (Thermo Fisher Scientific, Eugene, OR, USA).

### 4.3. Bisulfite Treatment

DNA from tissue and plasma samples was treated with an EZ DNA Methylation Kit by spin column (Zymo Research, Irvine, CA, USA) following the manufacturer’s protocol. The amount of DNA elution was 50 µL.

### 4.4. Droplet Digital PCR (ddPCR)

The presence of the *PDGFRA* D842V mutation and *SEPT9* hypermethylation were analyzed by a droplet digital PCR (ddPCR) assay using the QX200 Droplet Digital PCR System (Bio-Rad Laboratories, Hercules, CA, USA). Samples were prepared by mixing 10 μL of ddPCR Supermix for probes No dUTP (Bio-Rad Laboratories, Hercules, CA, USA), 1 μL of HindIII restriction enzyme (5 U/μL) (Thermo Fisher Scientific Baltics, Vilnius, Lithuania), 1 μL of FAM and HEX fluorescent probes (specific for mutant/methylated and wild-type gene, respectively), and 1 to 8.8 μL of template DNA in a final reaction volume of 20 μL. A total amount of 100 ng of tissue-derived DNA was added per well and a total amount between 10 and 50 ng of cfDNA from plasma was added per well. Three replicates were analyzed per sample. Water instead of DNA was used for no template control (NTC) and served as a control for detecting environmental contamination. Genomic DNA from WT tissue or WT plasma was used as a negative control to estimate the false-positive rate; a positive control containing genomic DNA from the tumor tissue was used to verify the assay performance and determine the threshold value of fluorescent signals. DNA from the HCT-116 cell line, which has hypermethylation of the *SEPT9* gene and *PDGFRA* D842V mutation, was used as positive control.

For detection of the *PDGFRA* D842V mutation, the commercially available assay ID dHsaMDV2516894, validated by Bio-Rad, was used (Bio-Rad Laboratories, Hercules, CA, USA) and the ddPCR cycles used for the *PDGFRA* gene were: 95 °C, 10 min, 40 cycles including 94 °C 30 s and 55 °C for 1 min for amplification, final cycle of 98 °C, 10 min and holding temperature at 4 °C, ∞.

For *SEPT9* hypermethylation analysis, primers for the *SEPT9* gene (5′-AGAGAATTTTGTTTGGTTGTTGTTTAAATATATAG-3′ and 5′-AAAAAAAAAATTCCTCCCCTTCC-3′) (Bio-Rad Laboratories, Hercules, CA, USA) and fluorescent probes for the methylated and unmethylated sequences (Methylated-FAM 5′-TGTAGAAGGATTTTGCGTTCGG-3′ Unmethylated-HEX 5′-TTGTAGAAG/ZEN/GATTTTGTGTGTTTGG-3′) were used once the DNA was converted with bisulfite. The design and validation of the primers were taken from Ma and colleagues [36]. The ddPCR cycles used were: 95 °C, 10 min, 40 cycles including 94 °C 30 s and 52 °C for 1 min for amplification, final cycle of 98 °C, 10 min and holding temperature at 4 °C, ∞.

Droplets were generated by a QX200 droplet generator (Bio-Rad Laboratories, Hercules, CA, USA) and end point PCR was performed on a T100 Thermal Cycler (Bio-Rad Laboratories, Hercules, CA, USA). After thermal cycling, the fluorescent signals of droplets were detected in the FAM and HEX channels of a QX200 droplet reader (Bio-Rad Laboratories, Hercules, CA, USA). The ddPCR data were analyzed using Quanta Soft v.1.7 Software (Bio-Rad Laboratories, Hercules, CA, USA). Results were reported as the number of copies of genetic alteration per μL/reaction. Poisson distribution was used to determine the concentration of *PDGFRA* D842V and *SEPT9* methylation.

### 4.5. Statistical Analysis

For the statistical analysis, the GraphPad Prism^®^ Version 8 software was used, and the results were reported in mean for each variable (concentration, ratio and abundance for each gene), standard deviation, and number of samples (*n*). To determine if a sample was positive for the *PDGFRA* D842V mutation, the concentration of *PDGFRA* D842V (copies/μL reaction) in the merged replicates of each sample was compared with a wild-type control (of similar WT concentration). To determine the *SEPT9* hypermethylation of samples (copies/μL reaction) in the merged replicates of each sample, they were compared with a wild-type control (of similar WT concentration) using a *t*-test, where the concentrations followed a normal distribution.

## 5. Conclusions

The findings from this study reveal that *PDGFRA* can be detected in GIST tissue down to 0.1 ng/μL using ddPCR. However, in early-stage disease, this detection may not be achievable through liquid biopsy. Instead, measures of *SEPT9* gene hypermethylation effectively differentiate between healthy subjects and GIST patients, with AUC values of 0.74, 0.77, and 0.79. An AUC of 0.79 indicates that hypermethylated *SEPT9* gene frequency is the most reliable measure. However, wide confidence intervals suggest the need for larger sample sizes in future studies to improve precision and statistical robustness.

## Figures and Tables

**Figure 1 ijms-25-06783-f001:**
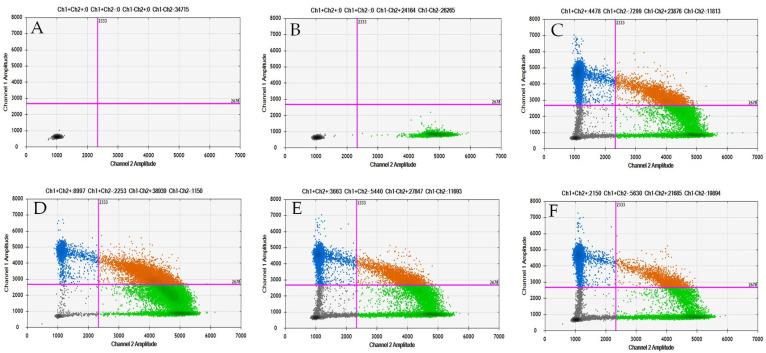
ddPCR results to detect the *PDGFRA* D842V mutation in DNA GIST tissue (100 ng) from patients at the Pathological Anatomy Service of Hospital Jiménez Díaz. Black dots indicate absence of both WT (wild type) and *PDGFRA* D842V mutation. Green dots indicate presence of WT *PDGFRA*. Blue dots indicate presence of *PDGFRA* D842V mutation. Orange dots indicate presence of both WT *PDGFRA* and *PDGFRA* D842V mutation. (**A**): NTC: no template control, (**B**): WT GIST tissue (AP 22155-805), (**C**–**F**): *PDGFRA* D842V GIST tissues, ((**C**): AP22155-802), ((**D**): AP22155-803), ((**E**): AP 22155-804), and ((**F**): AP 22155-806).

**Figure 2 ijms-25-06783-f002:**
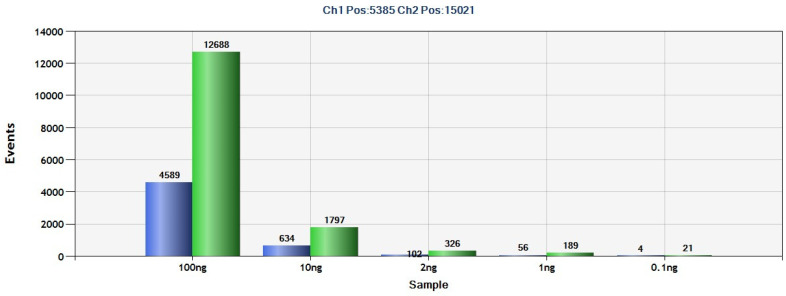
Number of *PDGFRA* D842V events (blue) and *PDGFRA* wild-type events (green) detected in DNA dilutions (100 ng, 10 ng, 2 ng, 1 ng, and 0.1 ng) extracted from a GIST tissue sample (AP 22155-803) of a patient at Hospital Jiménez Díaz, analyzed by ddPCR.

**Figure 3 ijms-25-06783-f003:**
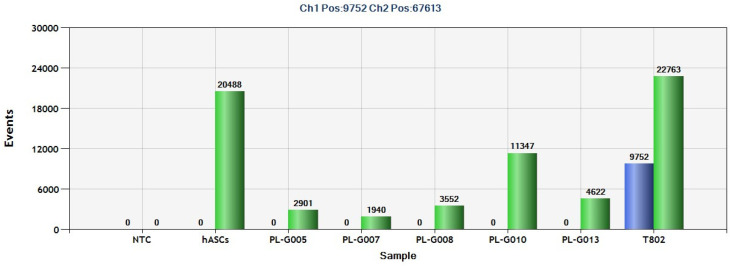
Number of *PDGFRA* D842V (blue) and *PDGFRA* wild-type (green) events in DNA (12 ng) isolated from plasma of GIST patients at Hospital Jiménez Díaz that were analyzed by ddPCR. NTC: no template control, hASCs: human Adipose Stem Cells as negative control (healthy cells), PL: plasmas (G005, G007, G008, and G013 from GIST patients with mutation *PDGFRA* D842V in tissue, and G010 with *PDGFRA* WT in tissue), T802: DNA from GIST Tissue (AP 22155-802) as positive control. Three replicates per sample.

**Figure 4 ijms-25-06783-f004:**
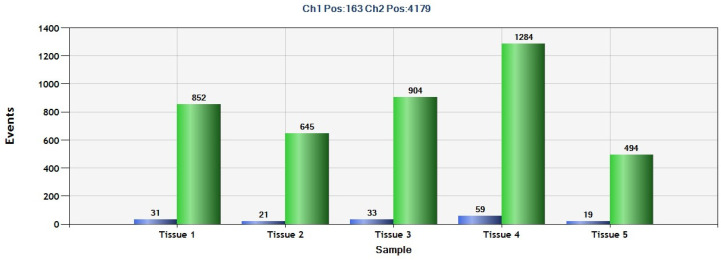
Number of events for *SEPT9* methylation (blue) and *SEPT9* WT (green) from GIST tissues (*n* = 5) of patients at Hospital Jiménez Díaz tested by ddPCR.

**Figure 5 ijms-25-06783-f005:**
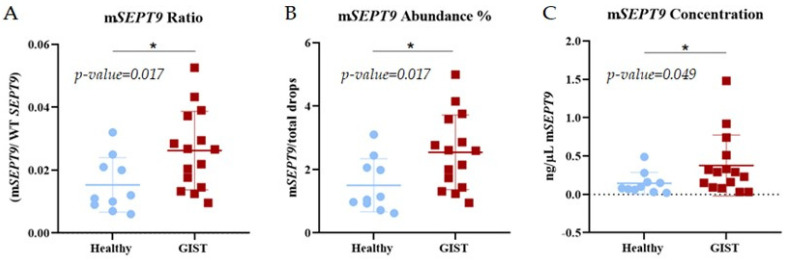
Comparison of *SEPT9* hypermethylation results in plasma between healthy participants *n* = 10 (Blue dots) and GIST patients *n* = 15 at Hospital Jiménez Díaz (Red dots) using *t*-test, α = 0.05. (**A**): Ratio of *SEPT9* methylated (concentration of *SEPT9* methylated/concentration *SEPT9*WT),* *p*-value: 0.0172, (**B**): Abundance of *SEPT9* methylated (*SEPT* methylated/*SEPT9*WT %), * *p*-value: 0.0170 and (**C**): Concentration of *SEPT9* methylated, ng DNA per µL of PCR reaction, * *p*-value: 0.0489.

**Table 1 ijms-25-06783-t001:** DNA alterations observed in patients with gastric tumors at Hospital Jiménez Díaz.

Code	DNA Alterations	Diagnosis	Age	Sex	Tumor Size (cm)	Tumor Stage
G001	N/A ^1^	Schwannoma	50	Female	5 × 4 × 3	N/A
G002	*KIT* exon 11 (W557R y V555E)	Gastric GIST	78	Male	3 × 3 × 2.5	pT2N0
G003	*KIT* exon 11 (W557R)	Gastric GIST	66	Female	4 × 4 × 3	pT2N0
G004	N/A	Fibromyxoma	59	Female	3 × 2 × 2	N/A ^1^
G005	*PDGFRA* exon 18 (D842V)	Gastric GIST	70	Male	2 × 2 × 2	pT2N0
G006	*PDGFRA* exon 18 (D842_S847 del)	Gastric GIST	78	Female	3 × 2 × 1	pT1N0
G007	*PDGFRA* exon 18 (D842V)	Gastric GIST	66	Male	3 × 2 × 2	pT2N0
G008	*PDGFRA* exon 18 (D842V)	Gastric GIST	71	Male	9 × 5 × 5	N/A
G009	*KIT* exon 11 (D579del)	Gastric GIST	70	Female	5 × 4 × 4	pT2N0
G010	No alterations in *KIT* and *PDGFRA*, WT	Gastric GIST	86	Female	4 × 3 × 3	pT2N0
G011	N/A	Lipoma	58	Female	5 × 4 × 3	N/A
G012	*KIT* exon 11 (W557_K558del)	Gastric GIST	54	Male	16 × 10 × 11	pT4N0
G013	*PDGFRA* exon 18 (D842V)	Gastric GIST	56	Male	4 × 3 × 3	pT2N0
G014	*KIT* exon 11 (V559_E561del)	Gastric GIST	52	Female	9 × 8 × 8	pT3N0
G015	*KIT* exon 11 (W557G)	Gastric GIST	68	Male	4 × 3 × 3	pT2N0
G016	*KIT* exon 11 (G556_K558delinsE)	Gastric GIST	62	Female	6 × 5 × 4	YpT3N0
G017	*KIT* exon 11 (M552_K558)	Gastric GIST	73	Female	4 × 3 × 2.5	ypT2yPN0
G018	*PDGFRA* exon 18 (D842V)	Gastric GIST	57	Male	5 × 5 × 1.5	pT2PN0

^1^ N/A: Not applicable.

**Table 2 ijms-25-06783-t002:** Anatomical pathological analysis of DNA extracted from GIST tissue. QC: Quality control.

Tissue Coded DNA	Alteration *PDGFRA*	% Tumoral	ng/µL DNA	260/280 QC DNA
AP 22155-802	*PDGFRA* exon 18 (D842V)	80	130	1.87
AP 22155-803	*PDGFRA* exon 18 (D842V)	60	138.2	1.9
AP 22155-804	*PDGFRA* exon 18 (D842V)	75	146	1.86
AP 22155-805	WT	85	176.2	1.9
AP 22155-806	*PDGFRA* exon 18 (D842V)	70	127.3	1.85

**Table 3 ijms-25-06783-t003:** Comparison of DNA concentration and *PDGFRA* WT events detected obtained from different plasma DNA isolation protocols to validate which protocol obtains more DNA.

Sample	ng/µL	WT *PDGFRA* Events Detected
Fresh plasma 3.5 mL	3.52	351
Cryopreserved plasma 3.5 mL	1.35	108.83
Large volume of cryopreserved plasma 12 mL	4.57	340
DNA concentrated by centrifugation	1.87	48.75
DNA concentrated by SpeedVacum	1.46	48.75

## Data Availability

Data is contained within the article.

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
