# Peer review of "Pilot Study by Liquid Biopsy in Gastrointestinal Stromal Tumors: Analysis of *PDGFRA* D842V Mutation and Hypermethylation of *SEPT9* Presence by Digital Droplet PCR"

_ijms, 2024, doi:10.3390/ijms25126783_

Round 1
Reviewer 1 Report
Comments and Suggestions for Authors
The current work is a preliminary study indicating the possibility of using droplet digital PCR in the diagnosis of gastrointestinal stromal tumors using liquid biopsy. The results obtained by the authors are promising, and the topic of the paper is clinically important. Minor errors in the paper do not affect my high evaluation of this work.
Line 32 - the abstract should be an autonomous part of the work. Therefore, it is necessary to expand all abbreviations used, such as AUC or CI.
Line 38 - keywords should be different from the title of the paper, in order to increase the visibility of the paper in databases.
Line 46 - either abdominal organs or organs of the gastrointestinal tract. GISTs are present throughout the gastrointestinal tract so it is logical that they are not present in other systems (of abdomen).
Line 211 - change to “87.5%”.
Line 329 - country of origin is missing for some laboratory items or reagents.
Line 379 - please describe how the authors designed and checked primers.
Author Response
Dear reviewer,
Thank you very much for your suggestions, we will try to answer one by one as clearly as possible:
The results obtained by the authors are promising, and the topic of the paper is clinically important. Minor errors in the paper do not affect my high evaluation of this work.
Response: Thank you for your valuable feedback. We appreciate your positive evaluation and have addressed the minor errors you noted as follows:
Line 32 - the abstract should be an autonomous part of the work. Therefore, it is necessary to expand all abbreviations used, such as AUC or CI.
Response: We expanded all abbreviations in the abstract to ensure it is fully autonomous. The revised sentence now reads: "Additionally, the results of Area Under the Curve (AUC) for the hypermethylated SEPT9 gene, analyzing concentration, ratio, and abundance were 0.74 (95% Confidence Interval (CI): 0.52 to 0.97), 0.77 (95% CI: 0.56 to 0.98), and 0.79 (95% CI: 0.59 to 0.99), respectively."
Line 38 - keywords should be different from the title of the paper, in order to increase the visibility of the paper in databases.
Response: To enhance the paper's visibility in databases, we have added additional keywords different from the title: "Rare disease, gastrointestinal stromal tumor (GIST), biomarkers, tissue biopsy, liquid biopsy, early diagnosis, cell-free DNA, droplet digital PCR (ddPCR), PDGFRA, hypermethylated SEPT9 gene."
Line 46 - either abdominal organs or organs of the gastrointestinal tract. GISTs are present throughout the gastrointestinal tract so it is logical that they are not present in other systems (of abdomen).
Response: We revised the sentence to clarify the most commonly affected organs by GISTs: "The most common organs affected by GISTs are those within the gastrointestinal tract, including the stomach (50%), small intestine (25%), rectum (5%), and esophagus."
Line 211 - change to “87.5%”
Response: We corrected the percentage to "87.5%." Thank you very much for this detail, which undoubtedly improves the quality of the manuscript.
Line 329 - country of origin is missing for some laboratory items or reagents.
Response: We reviewed and added the country of origin for all laboratory items and reagents. Thank you very much for this suggestion.
Line 379 - please describe how the authors designed and checked primers.
Response: We included a detailed description of how the authors designed and checked primers. The revised section now reads: "For detection of the PDGFRA D842V mutation, the commercially available assay ID dHsaMDV2516894, validated by Bio-Rad, was used (Bio-Rad Laboratories, Hercules, CA, USA). For SEPT9 hypermethylation analysis, methylation primers were validated in this article:
Ma, Z.Y.; Chan, C.S.Y.; Lau, K.S.; Ng, L.; Cheng, Y.Y.; Leung, W.K. Application of Droplet Digital Polymerase Chain Reaction of Plasma Methylated Septin 9 on Detection and Early Monitoring of Colorectal Cancer. Sci Rep 2021, 11, 23446, doi:10.1038/s41598-021-02879-8.
We added this information in the Materials and Methods section:
For SEPT9 hypermethylation analysis, primers for the SEPT9 gene (5'-AGAGAATTTTGTTTGGTTGTTGTTTAAATATATAG-3' and 5'-AAAAAAAAAATTCCTCCCCTTCC-3' (Bio-Rad Laboratories, Hercules, CA, USA)), fluorescent probes for the methylated and unmethylated sequence once the DNA is converted with bisulphite: Methylated-FAM 5'-TGTAGAAGGATTTTGCGTTCGG-3' Unmethylated-HEX 5'-TTGTAGAAG/ZEN/GATTTTGTGTGTTTGG-3 were used. The design and validation of the primers were taken from Ma and colleagues [37]."
Thank you again for your constructive feedback, which has helped improve the quality of our paper.
Reviewer 2 Report
Comments and Suggestions for Authors
The manuscript represents an exciting pilot study of liquid biopsy analysis. Despite being a pilot study, it is a valuable contribution to the field. Even though the number of samples is small, the rationale and conclusions are adequate. The methodology is suitable, although it can be improved as the authors suggested.
Author Response
Dear reviewer,
Thank you for your time in reviewing this manuscript.
The manuscript represents an exciting pilot study of liquid biopsy analysis. Despite being a pilot study, it is a valuable contribution to the field. Even though the number of samples is small, the rationale and conclusions are adequate. The methodology is suitable, although it can be improved as the authors suggested.
Response: Thank you for your encouraging feedback. We are pleased to hear that you find our pilot study valuable and that our rationale and conclusions are deemed adequate. While we acknowledge the small sample size, we appreciate your recognition of the study's contribution to the field. We also appreciate your comments on our methodology and will consider improvements as suggested.
Thank you for your constructive feedback.
Reviewer 3 Report
Comments and Suggestions for Authors
Dear Editor and Authors,
It was my pleasure to review this manuscript titled “Pilot Study by Liquid Biopsy in Gastrointestinal Stromal Tumors: Analysis of PDGFRA D842V Mutation and Hypermeth-ylation of SEPT9 Presence by ddPCR.” by Dr. Rocío Olivera-Salazar and his colleagues.
The premise of this pilot study is intriguing and focuses on applying liquid biopsy techniques to identify predominant mutations associated with GIST which are also therapeutic targets. In this way tissue biopsy may not be necessary or more potentially useful, tracking of tumor evolution and continuation of targeted therapy susceptibility.
One of the limitations of this study as much as I can see it is the utilization of tissue as opposed to blood to identify and validate the liquid biopsy technique. Given this is not the usual medium to be available for analysis and knowing that tissue contains a far superior sample reduces the validity of the results!! This is more so evident when plasma samples were used as opposed of diluted tissue samples from these GIST diagnosed patients which failed to detect the PDGFRA D842V mutation gene!! Despite utilizing plasma concentration techniques the authors were still unable to detect the mutation in liquid biopsy performed on plasma!! This is disappointing and in truth disproves the basic premise and hypothesis of the study but in science negative results are as valid (or sometimes even more) as positive ones!! Reporting these findings is fine as long as one does not downplay them or ignore them or try to make day into knight!!
Another limitation perceived is the number of samples utilized as no power analysis is performed to ensure a statistical meaningful result. This also needs mentioning!!
Finally, I have a question on when (in terms of time period and stage in the therapeutic process) were the plasma samples obtained? Was it before therapy? Was it at the time of tissue biopsy? Was it after? And if obtained before, how were they processed and stored and why since this was not a prospective study?
In general this is a well written and presented work. It flows and reads well and only some minor language editing is needed.
The introduction is well written providing a short overview of the pathology of GIST and then elucidating more on the genetic and molecular characteristics.
The methodology and the results are also well presented and illustrated and the discussion is thorough and ties things together nicely. Overall this is a good paper and I only have some very minor concerns about it. Kind regards to all.
Comments on the Quality of English LanguageMinor editing is needed.
Author Response
The premise of this pilot study is intriguing and focuses on applying liquid biopsy techniques to identify predominant mutations associated with GIST which are also therapeutic targets. In this way tissue biopsy may not be necessary or more potentially useful, tracking of tumor evolution and continuation of targeted therapy susceptibility.
Response:
Dear Reviewer,
Thank you for taking the time to review our manuscript. We appreciate your positive feedback on the intriguing premise of our pilot study and its focus on applying liquid biopsy techniques to identify predominant mutations associated with GIST, which are also therapeutic targets. We believe that this approach could potentially reduce the necessity for tissue biopsies and aid in tracking tumor evolution and therapy susceptibility.
One of the limitations of this study as much as I can see it is the utilization of tissue as opposed to blood to identify and validate the liquid biopsy technique. Given this is not the usual medium to be available for analysis and knowing that tissue contains a far superior sample reduces the validity of the results!! This is more so evident when plasma samples were used as opposed of diluted tissue samples from these GIST diagnosed patients which failed to detect the PDGFRA D842V mutation gene!! Despite utilizing plasma concentration techniques the authors were still unable to detect the mutation in liquid biopsy performed on plasma!! This is disappointing and in truth disproves the basic premise and hypothesis of the study but in science negative results are as valid (or sometimes even more) as positive ones!! Reporting these findings is fine as long as one does not downplay them or ignore them or try to make day into knight!!
Response: We acknowledge the limitation you pointed out regarding the use of tissue samples for validating the liquid biopsy technique. We chose to validate our study using tissue samples with ddPCR, despite the assay already being validated by Bio-RAD, to ensure that our experimental conditions could detect the PDGFRA D842V mutation effectively in a sample type with higher DNA concentration. This step was critical to confirm that our methodology was robust before applying it to plasma samples, which inherently contain less DNA. According to the literature (Fratte et al, 2020 and Serrano et al, 2020), this may be due to GIST had relatively low ctDNA shedding and mutations occurred at low allele frequencies; cfDNA was detected primarily in advanced disease.
Furthermore, we strongly agree with you that negative results are just as valid as positive ones, thank you very much for this insight.
Another limitation perceived is the number of samples utilized as no power analysis is performed to ensure a statistical meaningful result. This also needs mentioning!!
Despite conducting two statistical analyses (T-test and AUC) that yielded significant results, we agree with you that the number of samples is limited, especially because we are dealing with a rare disease. We have included this acknowledgment in the discussion section of the manuscript.
"Although statistical analysis showed significant results when comparing SEPT9 hypermethylation in healthy subjects with those with GIST, and the AUC also indicated significance, it is important to note that the limited number of samples, due to the rarity of the disease, is a limitation. To ensure a powerful analysis with statistically significant results, a larger sample size would be needed."
Thank you for highlighting this concern. Your feedback is valuable in ensuring the thoroughness and transparency of our study.
Finally, I have a question on when (in terms of time period and stage in the therapeutic process) were the plasma samples obtained? Was it before therapy? Was it at the time of tissue biopsy? Was it after? And if obtained before, how were they processed and stored and why since this was not a prospective study?
We apologize if we haven't been clear enough. To clarify, we have added the following statement: “Patients of both sexes, between 18 and 90 years old, with suspected GIST tumor were included in this study for the use of plasma samples. The plasma samples from patients suspected of having GIST were collected at the time of diagnosis, prior to surgery and patient selection was done after diagnosis provided by anatomical pathology at our hospital." Thank you for pointing this out.
The collection process was as follows: Plasma was isolated within 2 hours of blood collection following the conventional protocol for isolating DNA from plasma samples. The isolated samples were then stored at -80°C due to the large number required for ddPCR analysis, where each patient's sample occupied 3 wells in a 96-well plate (1 sample with 3 replicates). This necessitated the accumulation of plasma samples before commencing analysis.
In general this is a well written and presented work. It flows and reads well and only some minor language editing is needed.
The introduction is well written providing a short overview of the pathology of GIST and then elucidating more on the genetic and molecular characteristics.
The methodology and the results are also well presented and illustrated and the discussion is thorough and ties things together nicely. Overall this is a good paper and I only have some very minor concerns about it. Kind regards to all.
We appreciate your constructive feedback, which has been instrumental in enhancing the clarity and quality of our manuscript. We have strived to address your concerns comprehensively and transparently.
Thank you for your valuable input.
Kind regards,
Dr. Rocío Olivera-Salazar.
Reviewer 4 Report
Comments and Suggestions for Authors
The study titled "Pilot Study by Liquid Biopsy in Gastrointestinal Stromal Tumors: Analysis of PDGFRA D842V Mutation and Hypermethylation of SEPT9 Presence by ddPCR" advocates the use of droplet digital PCR (ddPCR) for the diagnosis of GIST in tissue samples and investigates its potential for early diagnosis by liquid biopsy, focusing on PDGFRA D842V mutation and SEPT9 hypermethylation. In this study, ddPCR was used to analyze the predominant PDGFRA mutation (D842V) in surgical tissue samples from 15 GIST patients and correlate the results with pathologists' diagnoses. The analysis was then extended to plasma samples to compare DNA changes between tumor tissue and plasma and also to investigate hypermethylation of the SEPT9 gene.
In the study, the PDGFRA D842V mutation was successfully detected in GIST tissue using ddPCR. However, despite various protocols to improve mutation detection in early disease stages, this remained a challenge, probably due to the low DNA concentration in plasma samples. In addition, the AUC results for the hypermethylated SEPT9 gene, where concentration, ratio and frequency were analyzed, were 0.74 (95% CI: 0.52 to 0.97), 0.77 (95% CI: 0.56 to 0.98) and 0.79 (95% CI: 0.59 to 0.99), respectively. These results suggest that the hypermethylated SEPT9 gene can serve as a biomarker for the early detection of GIST and is a promising tool in oncology to improve the quality of healthcare through earlier non-invasive diagnoses.
What were the specific criteria for the selection of patients and plasma samples for this study?
How were the DNA concentration protocols (e.g. use of fresh vs. cryopreserved plasma) optimized and validated?
How meaningful were the results regarding the detection of the PDGFRA D842V mutation in tissue samples compared to plasma samples?
Can you explain the statistical significance and relevance of the area under the curve (AUC) values obtained for SEPT9 hypermethylation?
What are the potential limitations and challenges of using ddPCR for the detection of early-stage GIST by liquid biopsy?
How does the heterogeneity of molecular profiles in GIST affect the detection rates and reliability of SEPT9 hypermethylation as a biomarker?
What are the implications of the results for the future clinical application of ddPCR in the diagnosis and monitoring of GIST?
What are the next steps for research based on the conclusions from this pilot study?
How can the results of the study be translated into clinical practice to improve early detection and treatment outcomes for GIST?
What specific challenges were encountered during the DNA extraction and ddPCR process and how were they overcome?
How do the results of this study compare with the existing literature on liquid biopsy and molecular testing in GIST?
What additional biomarkers or genetic alterations could be investigated in future studies to improve the early detection of GIST?
How could the relationship between hypermethylation of the SEPT9 gene and inflammation be further explored?
Author Response
Dear reviewer,
We have answered your questions one by one in the attached file.
We greatly appreciate your time in reviewing the manuscript, and we found your questions to be very pertinent and interesting,
Kind Regards,
Dr. Rocío Olivera-Salazar

Round 2
Reviewer 4 Report
Comments and Suggestions for Authors
The authors have performed appropriate alterations.